# Effects of Emotional Labor, Anger, and Work Engagement on Work-Life Balance of Mental Health Specialists Working in Mental Health Welfare Centers

**DOI:** 10.3390/ijerph20032353

**Published:** 2023-01-28

**Authors:** Kyung-Ok Lee, Kyoung-Sook Lee

**Affiliations:** 1Ulsan Nam-gu Mental Health Welfare Center, 3rd Floor, 132, Samsanjung-ro, Nam-gu, Ulsan 44698, Republic of Korea; 2Department of Nursing, University of Ulsan, 93 Daehak-ro, Nam-gu, Ulsan 44610, Republic of Korea

**Keywords:** work-life balance, quality of life, anger, policy

## Abstract

This study is a descriptive survey aiming to examine the general characteristics, emotional labor, anger, and work engagement of mental health specialists at mental health welfare centers and determine their effects on work-life balance (WLB). A total of 193 mental health specialists from 21 mental health welfare centers at metropolitan cities U and B were enrolled. A self-report and anonymous online questionnaire was used to collect data from 11 March to 1 April 2021. The collected data were analyzed using the *t*-test, analysis of variance, Scheffé test, Pearson’s correlation coefficients, and multiple regressions using SPSS Windows (Ver 25.0). We found that WLB is significantly negatively correlated with emotional labor (r = −0.47, *p* < 0.001), anger (r = −0.32, *p* < 0.001), and work engagement (r = 0.37, *p* < 0.001). The regression model confirmed that the male sex (β = 0.35, *p* = 0.002), moderate perceived health (β = −0.31, *p* = 0.003), poor perceived health (β = −0.35, *p* = 0.020), 1–3 years of career experience at a mental health welfare center (β = 0.27, *p* = 0.043), level of attentiveness required in emotional labor (β = −0.23, *p* = 0.014), and vigor of work engagement (β = 0.15, *p* = 0.005) were predictors of WLB, and these factors explained 43.1% of the variance. Supportive work policies and environments that promote perceived health, reduce emotional labor, and stimulate work engagement are needed to help mental health specialists at mental health welfare centers maintain a good WLB and enjoy a higher quality of life.

## 1. Introduction

Work-life balance (WLB) is one of the scales used to determine whether individuals live well in society today [1] and is aimed at helping individuals achieve life balance and spend time reasonably in handling various requests at work [2]. When workers attain a good WLB, their desire for growth is satisfied, they gain confidence and competency, and they become emotionally committed to their employing organizations [3]. In addition, WLB is socially important since it influences individual–organization harmony and affects work performance [4]. According to OECD’s database, Korean employees worked over 2000 h per year on average in 2017 [5]. Korean employees’ labor hours and productivity, as surveyed by the OECD, fall short of the expectations for adequate rest time and labor conditions to improve WLB [6]. Inappropriate WLB leads to greater emotional exhaustion and negative influence on work satisfaction. Long exposure to poor WLB causes various mental health problems and negatively affects mental health indexes [7]. WLB is very important for a healthy life. Mental health specialists working in mental health welfare centers who are always exposed to numerous civil service jobs in tense circumstances involving verbal and physical threats [8] seem to experience work-life imbalance.

Emotional labor refers to an individual’s process of trying to control and manage feelings that arise when their emotional expressions differ from those required by organizations [9]. Emotional labor is high among employees with many civil service jobs or with low positions [10] and among employees with work-life imbalance [11]. Mental health service specialists spend significant mental energy in the course of making interventions in their therapeutic relationships with clients, as well as in managing and controlling their feelings, thereby dealing with significant emotional labor [12]. The high emotional labor experienced by mental health specialists seems to influence their WLB.

Anger is a subjective feeling of antagonism. This feeling is defined by the emotional status of its components and the activation of the autonomic nervous system, triggering specific behavior patterns in individuals [13]. Expressions of anger are divided into anger-in, anger-out, and anger-control. Anger-in is an attempt to mitigate anger without showing reactions. Anger-out involves showing one’s anger by speaking or acting aggressively. Anger control involves using various strategies to control and alleviate anger appropriately in order to resolve the situation inducing anger [14]. Anger is often experienced by mental health specialists in mental health welfare centers. Given the nature of their job, expressing their anger is not always appropriate, and they mostly suppress the feeling instead [15]. Consequently, they developmental health problems like depression and more interpersonal problems [16]. Such individual and social negative influences create work-life imbalance [11].

Work engagement refers to a person’s positive and passionate mindset involving dedication and absorption in one’s job. People with high work engagement enjoy their jobs and tend to have a healthy and happy work status [17]. Work engage mental so affects mental health specialists’ WLB. Since high levels of emotional labor and anger have been proven to cause work-life imbalance [11], a strategy for maintaining WLB among mental health specialists in mental health welfare centers needs to be devised. However, the subjects of domestic-related studies on WLB were mostly employees in service areas, married female workers, and nurses [3,10,18,19]. Research on the WLB of mental health specialists is scarce.

In order to devise interventions for helping mental health specialists maintain WLB, the factors that affect WLB need to be identified. Therefore, this study investigates the effects of emotional labor, anger, and work engagement on the WLB of mental health specialists in mental health welfare centers and thereby obtains the fundamental data to improve mental health specialists’ quality of life by improving their WLB.

## 2. Methods

### 2.1. Study Design

This descriptive correlation survey aims to identify the general characteristics of mental health specialists working in mental health welfare centers, as well as to assess the effects of emotional labor, anger, and work engagement on WLB.

### 2.2. Study Subjects

The subjects of this study were mental health specialists working at 21 mental health welfare centers in metropolitan cities U and B. The number of samples for this study was determined using the G*Power 3.1 program (University of Dusseldorf, Düsseldorf, Germany), which indicated a sample size of 171 individuals based on a significance level of 0.05, power of 0.90, middle effect size of 0.15, and 15 predictor variables for regression analysis. The sample size was set at 188 subjects to account for a dropout rate of 10%. Out of 210 mental health specialists chosen in this study, 193 participated in the questionnaire survey. During the COVID-19 lockdown, participants filled out the questionnaires online.

The detailed acceptance criteria of the mental health welfare centers and participants are as follows: (1)Mental health specialists working at 21 mental health welfare centers in metropolitan cities like U. and B.; 2 metropolitan mental health welfare centers and 19 basic mental health welfare centers.(2)Mental health welfare center where the purpose of the study is understood and accepted by the head.(3)Mental health welfare centers with at least 15 employees, including mental health nurses, mental health social workers, and clinical psychologists.(4)About 60% of mental health specialists working at each center said they would respond to online surveys by providing online links.

All participants were from any of the twenty-one selected mental health welfare centers. They were fluent in the Korean language, able and willing to complete the questionnaire with granted informed consent.

### 2.3. Study Population

There was twenty-one mental health welfare centers that met the criteria. About 315 mental health specialists worked in the mental health welfare centers. The questionnaire was distributed to 60% of the mental health specialists—at each mental health welfare centers. 

### 2.4. Measures

#### 2.4.1. Emotional Labor

This study used the translated, modified, and supplemented emotional labor measurement scale developed by Kim M.J. [10] from the emotional labor scale developed by Morris and Feldman [9]. The subcategories of this scale are the frequency of emotional display, attentiveness to display rules required, and emotional dissonance [9]. Each subcategory has three questions, totaling nine questions for the scale [10]. Each question is based on a 5-point Likert scale ranging from “strongly agree”(5 points) to “strongly disagree” (1 point).The scores range from 9 to 45 points. The higher the score, the higher the emotional labor. The reliability of the original scale [9] had a Cronbach’s α of 0.80, whereas the revised scale [10] obtained a Cronbach’s α of 0.86. In the present study, the Cronbach’s α values of the entire scale, frequency of emotional display, attentiveness to display rules, and emotional dissonance were 0.80, 0.64, 0.67, and 0.78, respectively.

#### 2.4.2. Anger

The scale used for assessing anger in this study was the Korean version of the State-Trait Anger Expression Inventory(STAXI-K) translated by Kim G.H. [20] and standardized by Jeon et al. [21]. both of which modified on the STAXI developed by Spielberger [22]. The scale consists of three subcategories (anger-in, anger-out, and anger-control), each of which has eight questions for a total of 24 questions. Each question is based on a 4-point Likert scale from “strongly agree” (1 point) to “almost always agree” (4 points). The score for each dimension ranges from 8 to 32 points. A higher score indicates stronger anger-in, anger-out, or anger-control behaviors. At the time when the scale was developed, the anger-in subscale had a Cronbach’s α of 0.67, the anger-outsubscale 0.67, and the anger-control subscale 0.79. In the study by Jeon et al. [21], the Cronbach’s value for anger-in was 0.76, that for anger-out was 0.74, and that for anger-control was 0.85. In the study by Kim [20], the Cronbach’s value for anger-in was 0.71, that for anger-out was 0.78, and that for anger-control was 0.75. In the present study, the Cronbach’s α value for the entire scale, anger-out, anger-in, and anger-control were 0.68, 0.74, 0.78, and 0.72, respectively.

#### 2.4.3. Work Engagement

The work engagement scale used in this study was adopted from Baek [23], which was translated from the Dutch Utrecht Work Engagement Scale (UWES-9) developed by Schaufeli et al. [24]. The scale consists of nine questions: three questions each about vigor, dedication, and absorption. These questions are based on a 7-point Likert scale ranging from “strongly disagree” (1 point) to “strongly agree” (7 points). At the time the UWES-9 scale was developed, the Cronbach’s α value of the entire scale was 0.70, 0.68 for vigor, 0.91 for dedication, and 0.73 for absorption. For Baek’s scale [23], the Cronbach’s α value of the entire scale was 0.92, where as those for vigor, commitment, and absorption were 0.86, 0.89, and 0.80, respectively. In the present study, the Cronbach’s α value of the entire scale, vigor, dedication, and absorption were 0.90, 0.82, 0.85, and 0.79, respectively.

#### 2.4.4. Work-Life Balance

The WLB scale used in the present study was adopted from the scale Kim [3] modified and supplemented based on the study by Kim et al. [25]. The scale contains three categories, with five questions about work–family balance, five questions about work–leisure balance, and six questions about work–growth balance, for a total of 16 questions. The questions are scored using a 5-point Likert scale (1 = strongly disagree, 5 = strongly agree). The higher the score, the better the work-life balance. For the original scale [25], the Cronbach’s α for work–family balance, work–leisure balance, and work–growth balance were 0.67, 0.79, and 0.85, respectively. For the scale by Kim M.J. [3], the Cronbach’s α values were 0.84, 0.85, and 0.92, respectively. In the present study, the Cronbach’s α values were 0.68, 0.76, and 0.86, respectively, whereas that for the entire scale was 0.88.

### 2.5. Data Collection

For the ethical protection of the study participants’ rights, data were collected from 11 March to 1 April 2021, after approval by the institutional review board of U university(Approval Number: 1040968-A-2021-003). This researcher contacted mental health welfare centers located in metropolitan cities U and B; explained the purpose of the study and the method of data collection to their vice-director or team head; received their acceptance of request for cooperation; and finally obtained voluntary agreement to participate from the subjects who understood the purpose of this study. During the COVID-19 lockdown, web questionnaires were sent to representatives of the centers via e-mail. The subjects who agreed to participate were asked to complete the questionnaires individually, to fulfill the self-reporting attribute, and send them back. The study participants were assured that the survey content would not be used for purposes other than those specified for the study. Of 210 mental health specialists contacted, 193 completed the survey. A token worth 4000 KRW (Gifticon) was best owed to each of the survey participants. If a returned questionnaire had a missing item, it was considered incomplete and thus invalid for analysis. The data from all 193 participants were used for analysis.

### 2.6. Data Analysis

Data analysis was performed using SPSS for Windows (Ver. 25.0). The general characteristics of the participants were described using numbers and percentages. Emotional labor, anger, work engagement, and WLB were expressed as mean ± standard deviation. The differences in WLB scores according to the general characteristics were analyzed using the *t*-test and one-way analysis of variance after the normality test. The Scheffé test was performed as a post hoc test. The associations between emotional labor, anger, work engagement, and WLB were analyzed using Pearson’s correlation coefficients. The effects of emotional labor, anger, and work engagement on WLB were analyzed using multiple regression analysis.

## 3. Results

### 3.1. WLB Differences Based on the Subjects’ General Characteristics

A total of 193 individuals participated in the study, 153 (79.3%) of whom were women. Most of the participants were in their 30s (*n* = 114; 59.0%), and most did not practice a religion (*n* = 135; 69.9%). In terms of income, most participants earned 2.5 million won or less monthly numbered 96 (49.8%). Regarding subjective health status, most scored themselves as having a moderate health status (*n* = 110; 57.0%). Regarding job position, most participants worked as a team member (*n* = 137; 71.0%) for 1–3 years (*n* = 62; 32.1%) as a social worker (*n* = 144; 74.6%) at the mental health center. In terms of main responsibilities, most participants worked in suicide prevention (*n* = 59; 30.6%) and mental health promotion (*n* = 58; 30.1%). Of the participants, 93 (48.2%) had a turnover intention, whereas 100 (51.8%) did not.

The study subjects’ WLB varied significantly based on sex (t = 2.80, *p* = 0.006), subjective health status (F = 13.95, *p* < 0.001), job position (F = 3.57, *p* = 0.030), length of service at the mental health welfare center (F = 3.20, *p* = 0.014), type of occupation (F = 3.17, *p* = 0.044), and turnover intention (t = −3.86, *p* < 0.001). There was a difference in the WLB according to health status, and as a result of post-hoc analysis (Scheffe), subject who replied “Good” had a higher WLB than those who replied “moderate” or “Poor”. There was a difference in the WLB according to job position, and as a result of post-hoc analysis (Scheffe), subject who replied “Team member” had a higher WLB than those who replied “Team leader” or “Deputy director of center Standing team leader”. There was a difference in the WLB according to length of service at the mental health welfare center, and as a result of post-hoc analysis (Scheffe), subject who replied “1–3” had a higher WLB than subjects who answered “>10”. Detailed information is presented in Table 1.

### 3.2. Emotional Labor, Anger, Work Engagement, and WLB Scores

The results of the analysis of emotional labor, anger, work engagement, and WLB scores are presented in Table 2. The mean scores for emotional labor, frequency of emotional labor, attentiveness of emotional labor, and emotional dissonance were 3.21 ± 0.57, 3.60 ± 0.68l, 3.31 ± 0.60, and 2.73 ± 0.82, respectively.

With regard to their anger, the mean was 2.06 ± 0.22 of full four points. In its sub items, the mean for anger-out was 1.60 ± 0.34; the mean for anger-in 1.93 ± 0.45; the mean for anger-control 2.64 ± 0.40.

As for their work engagement, the mean was 4.25 ± 1.07 of full seven points. In its sub items, the mean for vigor was 3.45 ± 1.27; the mean for dedication 4.55 ± 1.28; the mean for absorption 4.74 ± 1.07.

With regard to their WLB, the mean was 3.03 ± 0.75 of full five points. In its sub items, the mean for work–family balance was 3.05 ± 0.79; the mean for work–leisure balance 3.03 ± 0.94; the mean for work–growth balance 3.02 ± 0.96.

### 3.3. Correlations between Variables

The correlations between emotional labor, anger, work engagement, and WLB are presented in Table 3. Emotional labor was positively correlated with anger (r = 0.48, *p* < 0.001) but negatively correlated with work engagement (r = −0.43, *p* < 0.001). Anger was also negatively correlated with work engagement (r = −0.20, *p* = 0.008). WLB was significantly negatively correlated with emotional labor (r = −0.47, *p* < 0.001) and anger (r = −0.32, *p* < 0.001) but was positively correlated with work engagement (r = 0.37, *p* < 0.001). Therefore, the lower the emotional labor and anger scores, the higher the work engagement and WLB.

### 3.4. Factors InfluencingWLB

Multiple regression analysis was conducted to determine the effects of emotional labor, anger, and work engagement on WLB. The results are presented in Table 4.

Univariate analysis revealed that sex, subjective health status, job position, length of service at the mental health welfare center, type of occupation, turnover intention, the sub variables of emotional labor (the frequency of emotional display, the attentiveness to display rules required, and emotional dissonance), sub variables of anger (anger-out, anger-in, and anger-control), and sub variables of work engagement (vigor, dedication, and absorption) were influential factors. Therefore, multiple regression analysis was conducted using these factors as predictor variables.

Among the predictor variables, ose measured in the nominal scale were processed as dummy variables.

The value obtained in the Durbin–Watson test was 1.90, which was close to 2, indicating no autocorrelation and, thus, residual independency. The value of multicollinearity (VIF) was between 1.21 and 5.25, which was less than 10, indicating no multicollinearity.

The entire regression model of the WLB was statistically significant (F = 7.31, *p* < 0.001). The multiple regression model analysis revealed males ex (β = 0.35, *p* = 0.002), moderate health status (β = −0.31, *p* = 0.003), poor health status (β = −0.35, *p* = 0.020), 1−3 years of service at the mental health welfare center (β = 0.27, *p* = 0.043), attentiveness to display rules required (β = −0.23, *p* = 0.014), and vigor in work engagement (β = 0.15, *p* = 0.005) as influential factors. The regression model for these factors obtained an explanatory power of 43.1%.

## 4. Discussion

This study verified that the influential factors on the WLB of mental health specialists were sex, subjective health status, length of service at the mental health welfare center, the attentiveness to display rules required, and vigor in work engagement. WLB was high among men, those with good subjective health status, those with 1–3 years of service, those with a low level of emotional labor, and those with a high level of work engagement. This study confirmed that sex was the biggest influential factor of WLB. Although little research involving mental health specialists has been conducted, a study involving radiologic technologists [26] and adult employees found that women experienced work-life imbalance more than men [27,28,29]. Therefore, female mental health specialists may require more support in terms of childcare and welfare and interest in order to raise their WLB.

Subjective health status was the second biggest influential factor on WLB. In other words, subject who replied “Good” had a higher WLB than those who replied “Moderate” or “Poor”. Therefore, various measures are needed to improve the subjective health status of mental health welfare center professionals. Generally, given the characteristics of mental health work, specialists will often suppress their emotions in handling situations involving civil service jobs [15]. This environment not only compromises their mental health, but also negatively influences their physical health, causing work-life imbalance. In order to maintain a good WLB, a strategy for improving employees’ overall health needs and a culture of coworker support needs to be established in the workplace [19]. In other words, a cultural facility or health promotion facility needs to be provided for health management, both for members of mental health welfare centers and mental health specialists, to improve physical and mental health. Furthermore, psychological support at work and a stable work environment can contribute toward mental health improvement.

In this study, length of service at the mental health welfare center was the third influential factor affecting WLB. Of the study subjects, 47% had worked 3 years or less. The fact that employees with more than 3years of service accounted for only 50% suggests that the work environment may be less than ideal. However, a report on employees at medical institutions [30] found that WLB did not different with length of service, which differed from the current results. The differences seemed to be caused by the differences in study subjects. Nevertheless, measures for increasing retention of mental health specialists at mental health service centers is important. To do this, improvements to human resources policies involving general measures for mental health need to be implemented; the value and system of related human resource curriculums need to be established [31]; and improvements to mental health specialists’ job satisfaction need to be met. These steps will not only increase the retention rate of mental health specialists in mental health welfare centers in such ways, they will also help them in providing and maintaining stable, high-quality services.

In this study, it was found that factors like age, religion, income, position at work, responsibilities, type of mental health occupation and turnover intention did not affect.A study on nurses reported that age [32] with a master’s degree or higher [33] also affected WLB, which disagreed with the findings of the present study. In a study of office workers, marital status, career, job position, duty, and income did not affect work-life balance [34]. Therefore, in the future, further study with various approaches to work and life balanceneed to be considered through follow-up studies.

In this study, emotional labor was another important influential factor on WLB, which agrees with that of a study on the impact of emotional labor on nurses’ WLB [11]. In offering their services, mental health specialists spend mental energy and experience emotional labor [12]. Many of them especially experienced severe tension and increased emotional labor while providing psychological support to self-quarantining individuals during the COVID-19 pandemic [33]. Therefore, in order to strike a good WLB, the emotional labor experienced by mental health specialists needs to be reduced. Emotional labor occurs among mental health specialists when they are unable to express their actual feelings or to express negative feelings they may feel toward their clients. Programs that can help them conserve psychological energy, encourage communication among employees at the mental health center, and reduce the attentiveness to display rules required can significantly improve mental health specialists’ WLB. In addition, the safety of work environments and the availability of organizational support need to be enhanced.

Work engagement was another important influential factor on WLB. The more vigor a person exerted into the job, the better the WLB. The present study found that vigor was lower than the levels of work engagement and absorption. This finding indicates that inspiring vigor among employees is essential, which can enhance their ability to deal with difficulties at work. A meta-analysis on work engagement [34] found that an intensive program for supporting individuals by providing job-related resources, leadership education, and health education exerted a small but critical improvement in work engagement. Helping improve vigor among mental health specialists to their increase work engagement can help them advance their WLB through the control of excess workload.

Among the factors examined in the present study, anger was not found to be influential on WLB. We found that the more anger-in and anger-out expressions were used, the greater the mental exhaustion [35,36]. A previous study showed that using anger-control as a desirable means of expressing anger resulted insignificantly lower mental health problems [22]. In the present study, the subjects’ anger-control levels were higher than the anger-in and anger-out levels because they used a desirable means of dealing with anger, which resulted in the absence of impact of anger on WLB.

This study identifies work-life balance influencing factors for mental health specialists at the mental health welfare center. Therefore, it is meaningful to provide basic data to improve their work-life balance. It is necessary to prepare a plan to pay more attention to areas of life other than work for mental health specialists at the mental health welfare center [34]. Promoting mental health specialists’ self-development and leisure activities cannot be simply achieved by reducing working hours. Additionally, workers who redused their working hours cannot engage in self-development or leisure activities due to reduced income [36]. Therefore, strategies and adequate policies are needed to find work-life balance through various methods, not just the reduction of working hours.

## 5. Conclusions

In conclusion, WLB was higher among mental health professionals like men with good subjective health conditions, people with one to three years of service, people with low emotional labor levels, and mental health professionals with high work participation levels. Therefore, mental health specialists, especially female ones, require opportunities and a conducive work environment for improving their subjective health status. In addition, policies that help reduce emotional labor should be implemented, which include programs that can help them deal with anger positively, encourage appropriate work engagement, and thereby maintain their WLB.

## 6. Limitation and Future Research

This study has several limitations. First, the study design was cross-sectional. This limits our confidence in determining the cause and effect in there relationships between the considered variables.

Second, mental health specialists working at 21 mental health welfare centers in two metropolitan cities were selected through convenience sampling. All the centers that participated in this study had a type of consignment operation. In fact, various management styles for mental health welfare centers are implemented, such as a nationally regulated operation. Therefore, care should be taken in generalizing the study results to all mental health specialists working in mental health welfare centers across the nation.

Third, during the COVID-19 pandemic, the survey was conducted as an anonymous, non-contact, self-administered online questionnaire survey. Since the researchers’ contact information was included in the questionnaire, participants were able to ask questions at any time. However, unlike a face-to-face questionnaire survey, this non-contact method was unable to establish real-time communication. Due to limitations in comprehension of the questionnaire, errors in measurement cannot be ruled out.

Fourth, this study did not include educational background, family support, organizational culture, atmosphere at work and leisure activities that could affect WLB.

Therefore, based on these limitations, this study makes a few suggestions. First, further research is needed on more diverse management styles of mental health centers involving more mental health specialists working in mental health welfare centers across the nation. In this way, fundamental data on policy development can be collected in order to establish a supportive work environment. Second, this study only made use of a questionnaire. Therefore, the assessment of physiological indexes in further studies would provide more quantitative measurements of the WLB of mental health specialists, especially mental health nurses. Third, based on the study results, a program to support the families, leisure, and growth of mental health specialists working in mental health welfare centers needs to be developed. Furthermore, establishing a work standard for mental health specialists can help them achieve a good WLB.

## Figures and Tables

**Table 1 ijerph-20-02353-t001:** Work-Life Balance Based on Participant Characteristics (*N* = 193).

Characteristics	Category	N	%	Work-Life Balance	
				Mean ± SD	Scheffé Test
Sex	Male	40	20.7	3.33 ± 0.78	2.80 (0.006)
	Female	153	79.3	2.95 ± 0.73	
Age	20s	49	25.4	3.14 ± 0.70	1.08 (0.343)
	30s	114	59.0	2.97 ± 0.77	
	≥40s	30	15.6	3.10 ± 0.78	
Religion	Yes	58	30.1	3.06 ± 0.84	0.30 (0.768)
	None	135	69.9	3.02 ± 0.72	
Income Per Month (million won)	<250	96	49.8	3.13 ± 0.67	2.21 (0.112)
	251–300	50	25.9	3.00 ± 0.80	
	>301	47	24.3	2.86 ± 0.84	
Subjective health status	Good ^a^	57	29.5	3.45 ± 0.82	13.95 (<0.001) a > b, c
	Moderate ^b^	110	57.0	2.88 ± 0.65	
	Poor ^c^	26	13.5	2.78 ± 0.69	
Position	Deputy director of center Standing team leader ^a^	15	7.8	2.90 ± 0.76	3.57 (0.030) a, b < c
	Team leader ^b^	41	21.2	2.78 ± 0.68	
	Team member ^c^	137	71.0	3.12 ± 0.76	
Length of service at the mental Health Welfare Center (years)	<1 ^a^	29	15.0	3.07 ± 0.54	3.20 (0.014) b > e
	1–3 ^b^	62	32.1	3.23 ± 0.76	
	4–5 ^c^	35	18.1	3.03 ± 0.69	
	6–10 ^d^	40	20.8	2.98 ± 0.84	
	>10 ^e^	27	14.0	2.62 ± 0.76	
Type of mental health occupation	Nurse ^a^	29	15.0	3.25 ± 0.70	3.17 (0.044) a > c
	Social worker ^b^	144	74.6	3.03 ± 0.74	
	Clinical psychologist ^c^	20	10.4	2.70 ± 0.83	
Main responsibilities	Management of severe mental patient	52	26.9	3.08 ± 0.78	0.23 (0.878)
	Mental health promotion	58	30.1	2.98 ± 0.70	
	Suicide prevention	59	30.6	3.06 ± 0.79	
	Mental health promotion in children and adolescents	24	12.4	2.99 ± 0.77	
Turnover Intention	Yes	93	48.2	2.82 ± 0.74	−3.86 (<0.001)
	No	100	51.8	3.23 ± 0.71	

a–e exist as indication for post-hoc analysis.

**Table 2 ijerph-20-02353-t002:** Scores for Emotional Labor, Anger, Work Engagement, and Work-Life Balance.

Variable Subcategories	Range	Mean ± SD	Min	Max
Emotional Labor	1–5	3.21 ± 0.57	2	5
Frequency of emotional display		3.60 ± 0.68	2	5
Attentiveness to display rules required		3.31 ± 0.60	2	5
Emotional dissonance		2.73 ± 0.82	1	5
Anger	1–4	2.06 ± 0.22	2	3
Anger-out		1.60 ± 0.34	1	3
Anger-in		1.93 ± 0.45	1	3
Anger-control		2.64 ± 0.40	2	4
Work Engagement	1–7	4.25 ± 1.07	1	7
Vigor		3.45 ± 1.27	1	7
Dedication		4.55 ± 1.28	1	7
Absorption		4.74 ± 1.07	1	7
Work-Life Balance	1–5	3.03 ± 0.75	1	5
Work–family balance		3.05 ± 0.79	1	5
Work–leisure balance		3.03 ± 0.94	1	5
Work–growth balance		3.02 ± 0.96	1	5

**Table 3 ijerph-20-02353-t003:** Correlations among Emotional Labor, Anger, Work Engagement, and Work-Life Balance.

Variable	Emotional Labor	Anger	Work Engagement	Work-Life Balance
	r(p)	r(p)	r(p)	r(p)
Emotional Labor	1			
Anger	0.48 (<0.001)	1		
Work Engagement	−0.43 (<0.001)	−0.20 (0.008)	1	
Work-Life Balance	−0.47 (<0.001)	−0.32 (<0.001)	0.37 (<0.001)	1

**Table 4 ijerph-20-02353-t004:** Factors Influencing Work-Life Balance.

		Work-Life Balance					
Variable		B	SE	β	t	*p*	VIF
(Constant)		4.36	0.56		7.74	0.000	
Sex	Female (ref)						
	Male	0.35	0.11	0.19	3.10	0.002	1.21
	Good (ref)						
Subjective Health Status	Moderate	−0.31	0.10	−0.21	−3.06	0.003	1.52
	Poor	−0.35	0.15	−0.16	−2.34	0.020	1.55
Position	Deputy director of center (ref) Standing team leader						
	Team leader	0.09	0.19	0.05	0.47	0.636	3.67
	Team member	0.39	0.21	0.23	1.88	0.062	5.25
Length of Service at the Mental	<1 (ref)						
Health Welfare Center (Years)	1–3	0.27	0.13	0.17	2.04	0.043	2.31
	4–6	0.28	0.16	0.14	1.75	0.081	2.15
	6–10	0.10	0.19	0.05	0.52	0.606	3.43
	>10	−0.11	0.22	−0.05	−0.51	0.614	3.54
Type of Mental Health Occupation	Nurse (ref)						
	Social worker	−0.14	0.13	−0.08	−1.05	0.295	1.87
	Clinical psychologist	−0.17	0.18	−0.07	−0.94	0.351	1.81
Turnover Intention	Yes (ref)						
	No	−0.03	0.10	−0.02	−0.26	0.796	1.52
Emotional Labor	Frequency of emotional display	−0.07	0.08	−0.06	−0.84	0.405	1.82
	Attentiveness to display rules required	−0.23	0.09	−0.18	−2.48	0.014	1.80
	Emotional dissonance	−0.11	0.07	−0.12	−1.59	0.115	1.84
Anger	Anger-out	0.05	0.14	0.02	0.38	0.702	1.38
	Anger-in	−0.22	0.12	−0.13	−1.86	0.065	1.64
	Anger-control	−0.14	0.13	−0.08	−1.11	0.269	1.58
Work Engagement	Vigor	0.15	0.05	0.25	2.84	0.005	2.54
	Dedication	0.02	0.06	0.03	0.36	0.722	3.00
	Absorption	−0.02	0.06	−0.02	−0.25	0.800	2.58

R^2^ = 0.499, ad R^2^ = 0.431, F = 7.31 (*p* < 0.001), DW = 1.90.

## Data Availability

Data supporting reported results can be requested from the first author (kolee0905@hanmail.net).

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
