# Peer review of "Effects of Emotional Labor, Anger, and Work Engagement on Work-Life Balance of Mental Health Specialists Working in Mental Health Welfare Centers"

_ijerph, 2023, doi:10.3390/ijerph20032353_

Round 1

Reviewer 1 Report

The reviewed paper takes up a considerable and up-to-date research problem.

The article “investigates the effects of emotional labor, anger, and work engagement on the WLB of mental health specialists in mental health welfare centers and thereby obtains the fundamental data to improve mental health specialists’ quality of life by improving their WLB”.

The set goal was achieved.

It is worth emphasizing the rich empirical achievements in the discussed area. In the reviewed article, the process of argumentation was carried out in an understandable, logical, and specific manner, while at the same time demonstrating the importance of the results of scientific research in a comparative manner.

The literature review of the subject includes the latest literature on the subject, but some of the root articles are also implemented.

I propose to enrich the content of the article with further items of literature on the subject.

I suggest expanding the concluding remarks because they should be more comprehensive.

Furthermore, I suggest checking certain aspects of the bibliographic reference lists. For example, the name of the journal must be written with its abbreviation, the volume must be written in italics and the year in bold, as well as text should be justified. There are a few shortcomings. It requires improvements.

Author Response

Discussion

  1. I propose to enrich the content of the article with further items of literature on the subject.

Thank you for the thoughtful review. I revised the discussion by referring to more literature. The revised part is marked in red and is shown below.

 Subjective health status was the second biggest influential factor on WLB. In other words, subject who replied “Good” had a higher WLB than those who replied “Moderate” or “Poor”. Therefore, various measures are needed to improve the subjective health status of mental health welfare center professionals.

In this study, it was found that factors like age, religion, income, position at work, responsibilities, type of mental health occupation and turnover intention did not affect. A study on nurses reported that age[32] with a master's degree or higher[33] also affected WLB, which disagreed with the findings of the present study. In a study of office workers, martial status, career, job position, duty, and income did not affect work-life balance[34].Therefore, in the future, further study with various approaches to work and life balance need to be considered through follow-up studies.

This study identifies work-life balance influencing factors for mental health specialists at the mental health welfare center. Therefore, it is meaningful to provide basic data to improve their work-life balance. It is necessary to prepare a plan to pay more attention to areas of life other than work for mental health specialists at the mental health welfare center [34]. Promoting mental health specialists’ self-development and leisure activities cannot be simply achieved by reducing working hours. Additionally, workers who redused their working hours cannot engage in self-development or leisure activities due to reduced income [39]. Therefore, strategies and adequate policies are needed to find work-life balance through various methods, not just the reduction of working hours.

Conclusions

  1. I suggest expanding the concluding remarks because they should be more

comprehensive.

In conclusion, WLB was higher among mental health professionals like men with good subjective health conditions, people with one to three years of service, people with low emotional labor levels, and mental health professionals with high work participation levels.

References

I checked and corrected.

  1. Furthermore, I suggest checking certain aspects of the bibliographic reference lists. For example, the name of the journal must be written with its abbreviation, the volume must be written in italics and the year in bold, as well as text should be justified. There are a few shortcomings. It requires improvements.

I checked and corrected.

Reviewer 2 Report

First I would like to congratulate the authors for choosing this theme. It is one of the very important themes of the mental health of a working population. The study as a whole makes a very good impression.

Keywords should be rephrased, most of them existed in the title.

The introduction is clearly written and led to the core of the problem. Maybe the authors could give data about a similar study from the literature.

In the Method, the authors said that out of 210 mental health specialists were chosen. Please explain better how they are chosen...The authors explain that they used inclusion criteria, but it's not clear does 210 specialist is all of the specialists which have fulfilled the criteria or if they have chosen 210 from a larger sample. In the  Discussion and the part of the limitation of the study, the authors wrote it is convenience sample.

In the Data analysis exist misspelling -" smean". I think it is mean.

The limitations of the study also need to be more detailed. For example...does the design of the study also represent one of the limitations. In addition, the authors used the words predictors...does this type of study could give information about predictors or not.

Author Response

Keywords should be rephrased, most of them existed in the title. : Changed to MeSH Key word.

Keywords: Work–life balance, Quality of life, Anger, Policy

In the Method, the authors said that out of 210 mental health specialists were chosen. Please explain better how they are chosen...The authors explain that they used inclusion criteria, but it's not clear does 210 specialist is all of the specialists which have fulfilled the criteria or if they have chosen 210 from a larger sample. In the  Discussion and the part of the limitation of the study, the authors wrote it is convenience sample.

The detailed acceptance criteria of the mental health welfare centers and participants are as follows:

1) mental health specialists working at 21 mental health welfare centers in metropolitan cities like U. and B. ; 2 metropolitan mental health welfare centers and 19 basic mental health welfare centers.

2) mental health welfare center where the purpose of the study is understood and accepted by the head.

3) mental health welfare centers with at least 15 employees, including mental health nurses, mental health social workers, and clinical psychologists.

4) about 60% of mental health specialists working at each center said they would respond to online surveys by providing online links.

All participants were from any of the twenty-one selected mental health welfare centers. They were fluent in the Korean language, able and willing to complete the questionnaire with granted informed consent.

Study Population

There was twentyone mental health welfare centers that met the criteria. About 315 mental health specialists worked in the mental health welfare centers. The questionnaire was distributed to 60% of the mental health specialists - at each mental health welfare centers.

In the Data analysis exist misspelling -" smean". I think it is mean.

I checked, " I modified "smean" to "mean".

The limitations of the study also need to be more detailed. For example...does the design of the study also represent one of the limitations. In addition, the authors used the words predictors...does this type of study could give information about predictors or not.

  1. Limitation and Future Research

This study has several limitations. First, the study design was cross-sectional. This limits our confidence in determining the cause and effect in there relationships between the considered variables.

Fourth, this study did not include educational background, family support, organizational culture, atmosphere at work and leisure activities that could affect WLB.